# Influence of Load Plates Diameters, Shapes of Columns and Columns Spacing on Results of Load Plate Tests of Columns Formed by Dynamic Replacement

**DOI:** 10.3390/s21144868

**Published:** 2021-07-16

**Authors:** Sławomir Kwiecień

**Affiliations:** Faculty of Civil Engineering, Silesian University of Technology, 44-100 Gliwice, Poland; slawomir.kwiecien@polsl.pl

**Keywords:** ground improvement, dynamic replacement, stone columns, load plate test, numerical analysis, FEA

## Abstract

The dynamic replacement method is used to strengthen the subgrade of objects, usually up to 5 to 6 m thick. After the improvement process, acceptance tests in the form of load testing are carried out. Interpretation of the test results can cause some difficulties. Dynamic replacement results in a situation where columns of different shapes, loaded with plates of diameters usually smaller than the head diameter and in the vicinity of adjacent columns, are subjected to load tests. In order to demonstrate the influence of these factors, a spatial model of soil strengthened by dynamic replacement, comprising four material zones, was calibrated on the basis of load testing. The following models were used in the analysis: linear-elastic, elastic–perfectly plastic (Coulomb–Mohr) and elastic–plastic with isotropic hardening (Modified Cam-Clay). This formed the basis for 105 numerical models, which took into account the actual shapes of the columns made at various spacings, subjected to load tests with plates of various diameters. The analyses of the settlements, calculated moduli and stress distribution in the loaded system showed how the results were significantly influenced by mentioned factors. This implies that the interpretation of the results of load tests should be based on advanced spatial numerical analyses, using appropriate constitutive models and including the considered factors.

## 1. Introduction

When designing the foundation for a structure, the suitability of the building subgrade is assessed qualitatively and quantitatively. Under the qualitative assessment, it is possible to identify “weak” soils (e.g., soft cohesive soils and organic soils) based on a review of the geotechnical documentation. The quantitative assessment checks the conditions of the ultimate limit states and serviceability limit states. In the case of a negative quantitative assessment, the object can be founded on a subgrade improved by geotechnical engineering methods. This includes several dozens of methods used to improve the soil parameters. In the case of weak layers occurring in the soil in the form of cohesive, organic or human-made soils, with a typical thickness of up to 5 to 6 m [1] and the maximum of 8 m [2], it is possible to strengthen them by using the dynamic replacement method.

The dynamic replacement method, which constitutes a development and also a complement of the dynamic compaction and dynamic consolidation methods, was first applied in 1975 [3]. The principle of dynamic replacement is to drop a heavy tamper and fill the displaced soil with an aggregate to form columns (Figure 1) of material which vary in grain size (from sand subfractions to boulders) [4]. After column formation, changes occur in the surrounding soil. They depend on the distance from the column, elapsed time, and the type and the initial condition of the soil. During the strengthening process, the soil softens in close vicinity of the column, but then soil parameters increase over time [5]. Some authors have observed the strengthening of the surrounding soil [6,7]. The dynamic replacement method increases the bearing capacity, reduces settlement and speeds up the consolidation process of improved soil [3,8].

The dynamic replacement designer’s task is to determine the grid of columns, including their diameters, spacing and lengths.

After the subsoil is strengthened by dynamic replacement, acceptance tests are carried out to check the bearing capacity and stiffness requirements for a single column, in the form of load tests [1,3,9].

For tests of up to several hundred kilopascals (e.g., 0–300 kPa), the loading plate is placed on the compacted and leveled head of the column, and the loading is carried out by crane-locked actuators used for ground improvement. Settlement is measured by using three or four mechanical or electronic sensors supported by independent steel frames spaced evenly around the slab. The tests are carried out by using the method of constant load steps, with the adopted value of stabilization of settlements, including the range of primary and secondary pressures. This concept is used in daily acceptance practice.

In order to achieve higher pressures, it is necessary to build a test rig similar to the one used for pile load tests (a set of steel beams against which the actuators are locked, anchored to the ground, e.g., with piles).

This gives load–settlement ratios within in the range of primary or primary and secondary pressures, allowing for the determination of the deformation modulus of the improved soil or the bearing capacity of the columns. For the former, solutions of the Boussinesq problem for an elastic half-space are used, with the deformation modulus (E) determined according to Formula (1):E = qwB(1−*υ*^2^)/s,(1)
where q is the load, w is the influence factor depending partly on the shape of the loading area and its stiffness, B is the width or diameter of the loaded area, *υ* is the Poisson ratio and s is the recorded settlement.

The column bearing capacity can be determined by specifying the column displacement limits, e.g., 10% of the head diameter (0.1D_h_).

The characteristics of dynamic replacement, especially the variability of the obtained diameters of the column heads (e.g., 1.4–4 m [4] and 2.5–5 m [2]) makes it necessary to perform the load tests at various ratios of the loading plate diameter (D_p_) to column head diameter (D_h_), usually less than 1. The reason for using smaller plate diameters is also due to the funds needed for the construction of heavy duty stations and the time required to construct them. Furthermore, dynamic replacement columns can be characterized by different shapes, often non-cylindrical, and based on layers of different stiffness [10]. The use of plates in load tests for which the D_p_/D_h_ ratio is variable may result in an area having different volumes. In addition to the loaded column and its immediate surroundings, it may also include the column padding layer and even adjacent columns. This, in turn, will affect the results, e.g., settlements and column-bearing capacities. An example of this involves the results of three load tests performed by the author, on three columns formed in similar soil conditions, with plates for which the diameter of load was 1.2 m and D_p_/D_h_ ratio was variable, at 0.55–0.75 [10]. It is clear that at higher D_p_/D_h_ ratios the system experiences larger vertical displacements (Figure 2a) during testing, and, for example, the primary deformation moduli calculated according to Formula (1) (B = 1.2 m, w = 0.75 (circular shape of stiff plate), *υ* = 0.23) become smaller (Figure 2b).

The abovementioned considerations prompted the author to examine these issues. This paper determines the influence of these factors on the results of the load tests, i.e., loading plate diameter and column shapes; how they are founded; and the presence of adjacent columns. The basis of the analysis is the FEM numerical model, calibrated on the basis of field load tests and laboratory test. A similar approach is used in practice [8]. It facilitated the definition of four material zones in the form of a column, weak soil surrounding the column, high-strength and stiff soil lying under the column, and the loading plate. The analyses involved columns of seven different characteristic shapes obtained from studies in monograph [10] supplemented by additional excavations, formed as singles and at 1.5D_h_ and 2D_h_ spacing, loaded with plates ranging from 0.2D_h_ to 1.0D_h_. The load–settlement relationship was analyzed for each model, with a particular focus on the possibility of loss of bearing capacity of the tested column; the deformation modules of the improved soil were determined, and the distributions of horizontal and vertical stresses and the direction of the principal stress distribution were analyzed.

## 2. Materials and Methods

### 2.1. Calibration and Verification of Numerical FEM Model for Soil Strengthened by Dynamic Replacement

The basis for model calibration was a dynamic replacement column load capacity test carried out on the construction site for a road in Poland. The strengthened layer consisted of firm clayey mud approximately 3 m thick, underlain by medium dense gravels with the addition of cobbles and rock layers. In the section under consideration, the dynamic replacement was performed using an equilateral triangle grid with 3 m sides, made from crushed sandstone in the 0–400 mm fraction with a pounder in the shape of a truncated cone with the diameters: bottom 1 m, top 1.2 m, height 1.8 m and weight 11.5 Mg, dropped from a height up to 13.5 m. The column to be tested was formed from 13.65 m_3_ of material and 16 pounder drops. The surveyed column head diameter was 2.2 m [11,12].

The view of the load test stand is shown in Figure 3.

The load test was carried out by using the constant load-step method. Each step was maintained until the column settlement rate exceeded 0.05 mm/15 min. Loads were applied by using three hydraulic actuators with a range of 0–1300 kN, positioned on a 1.2 m diameter plate (ratio D_p_/D_h_ = 0.55). The settlement was measured by using three electronic sensors with a range of 0–100 mm and readout accuracy of 0.01 mm, spaced circumferentially at 120°. For a final pressure of 1373 kPa, the column settlement was about 68 mm. More detailed results of the load tests are presented in the verification section of the proposed numerical model in Figure 4.

After completing the bearing capacity test, the column shape was surveyed (excavation with a backhoe loader) and samples of the soil (Category A, Sample quality class 1 [13]) surrounding the column (clayey mud), from a depth of about 1.5 m, were taken for further laboratory tests.

Physical and mechanical parameters were determined in accordance with the Polish standard PN-88/B-04481 [14].

To determine the mechanical parameters, oedometric tests and triaxial tests were carried out.

Oedometric tests were carried out in the load range of 0–400 kPa, in the following cycles: primary loading (0–200 kPa), unloading (200–12.5 kPa) and reloading (12.5–400 kPa).

Triaxial tests were conducted with consolidation and drainage.

Oedometric and triaxial tests were mainly carried out to calibrate parameters of the Modified Cam-Clay model.

A detailed description of the tests and its results were presented in the Ph.D. thesis [15] and monograph [4].

The tested column had a barrel-like shape (Figure 4). The base and head diameters were 2.2 m, and the column length was 2.7 m. The column was formed from a layer of medium compacted gravels [11,12].

The results from the laboratory tests of the physical and mechanical characteristics of the weak soil surrounding the column are presented in Table 1.

Calibration and verification of the proposed model of the soil strengthened by dynamic replacement was carried out in Z_Soil ver. 11.15 [16].

A spatial model was adopted for the analysis, which included both the test-loaded column and the adjacent columns with their shapes determined by excavation (Figure 4). The symmetry of the system is accounted for in the model. Standard geotechnical boundary conditions were applied—horizontal and vertical displacements were blocked at the bottom edge of the system, while horizontal displacements were blocked at the lateral vertical edges. Since the calculations were carried out by taking into account the filtration of water in the soil pores at the upper edge of the model, the filtration boundary condition was introduced in the form of zero pore water-pressure values [16].

The layout of the discrete model (without boundary conditions) is shown in Figure 4.

FEM computational analysis concerning the issue of strengthening cohesive (organic) soils with granular material (dynamic replacement columns) suggests the use of specific constitutive soil models in the computational analysis. In the case of granular soils forming both the column and the underlying soil layer (gravel with the addition of pebbles), characterized by low compressibility, the elastic–perfectly plastic model with a Coulomb–Mohr yield surface and non-associated flow rule can be successfully applied. During column loading, plastic deformation will dominate in large areas of weak soil; hence, the models justified for these soils are those with plastic strengthening and weakening combined with porosity changes. This prefers critical state models, particularly the well-known and implemented Modified Cam-Clay model in FEM programs.

The calibration of the selected constitutive models was carried out by using the semi-inverse method. The parameters of the selected models were determined based on laboratory tests, geotechnical documentation, the literature, standards and FEM analysis. The criterion for FEM calibration was obtaining correspondence between the numerical settlements and those determined during the field tests.

The parameters of the linear-elastic model simulating the operation of the concrete loading plate were adopted from PN-EN 1992-1-1:2008 as for C12/15 concrete [17].

In case of gravels, the parameters of elastic–perfectly plastic model with a Coulomb–Mohr yield surface were adopted on the basis of geotechnical documentation.

The model parameters for the strengthened soil simulated with the Modified Cam-Clay isotropic reinforcement critical state model were selected based on oedometric (λ and κ) [4,15] and triaxial (M) tests [4,15], and from the literature (υ) [18]. The porosity index (e_0_) and filtration coefficient (k_v_ = k_h_) were determined by laboratory tests (Table 1).

Some of the parameters of the elastic–perfectly plastic model with a Coulomb–Mohr yield surface (E, ϕ, ψ, c), which simulates the operation of dynamic replacement columns, were determined on the basis of semi-inverse FEM analysis. The filtration coefficient (k_v_ = k_h_) and Poisson’s ratio (υ) of the column material were determined from the literature [19].

FEM analysis made it possible to determine the parameters of the model simulating the work of columns, i.e., E = 60 MPa, ϕ = 46°, c = 5 kPa, ψ = 20°, for which the modified coefficient of determination was 0.989. Small value of column cohesion is probably due to the interlocking of column particles (made from crushed sandstone) phenomenon [20]. The results of the back analysis, along with the bearing capacity study, are shown in Figure 5.

A summary of the constitutive models and their parameters is shown in Table 2.

### 2.2. Classification of Dynamic Replacement Columns by Shape

The author has been interpreting the column shapes formed by the dynamic replacement method since 2007. This paper continues the work presented in References [4,10]. A total of 65 columns were surveyed in different ground conditions and during different road and retaining engineering construction projects. Each of the surveyed columns was unearthed with a backhoe loader. The column diameter and length were measured and complemented with photographs from the terrain level. The ground conditions were determined from the geotechnical survey reports compiled for the construction projects, and the reports were refined during the test digs.

All of the surveyed and formed columns were formed with pounders 1 to 1.6 m wide, 1.2 to 2 m high and 10–24 Mg in weight, dropped onto the soil from a height of up to 25 m.

The survey and tests facilitated the classification of the column shapes into two primary groups. Group I included end bearing columns (with flat bases). Group II included floating columns (with semi-circular bases). Group I featured four subgroups. Group II featured three subgroups. The subgroup classification criterion was the column diameter change along the column length.

For the end bearing columns in Group I, the columns’ shapes and diameters were primarily determined by the depth of the strengthened soil. In this group, the columns had fixed diameters along their length (Subgroup Ia), with maximum diameters at the base (Ib), at the mid-length point (Ic) and at the column bottom part (Id), respectively. For the floating columns, three specific column shapes were identified: cylindrical (Subgroup IIa), wide-base (Subgroup IIb) and wide-head (Subgroup IIc).

Classification of the investigated dynamic replacement columns is presented in Table 3.

A summary of the variation of column diameters in relation to their head diameters (mean values), for Groups I and II, is presented in Figure 6 and Figure 7. It could help in economic design of dynamic replacement columns.

### 2.3. Calculation Assumptions

The numerical FEM model calibrated in Section 2.1 (adopted material zones and constitutive models with parameters) and the determined variation of diameters of the columns in Groups I and II (Section 2.2) were the basis for the calculations of the effects of selected factors on the results of the load tests, i.e., loading plate diameter, column shape and how they are founded, and the occurrence of adjacent columns.

The column shapes described by the seven subgroups were mapped in the calculations by adopting column head diameters (D_h_) equal to the mean values, i.e., 1.93 m (Ia), 1.63 m (Ib), 2.0 m (Ic), 1.85 m (Id), 1.99 m (IIa), 1.73 m (IIb) and 2.59 m (IIc).

Group I column lengths (based on high stiffness layers) were assumed to be equal to the limit values for each subgroup, i.e., 2.0 m (Ia), 2.7 m (Ib), 3.8 m (Ic) and 4.4 m (Id). For Group II columns, the modeled column lengths corresponded to the largest lengths in each subgroup, at 4.5 m (IIa), 4.2 m (IIb) and 4.4 m (IIc). The Group II columns were underlain by a layer of weak soil with a thickness of 0.5 m.

The height of each model was equal to the thickness of the layer strengthened by dynamic replacement plus 2 m of the underlying subgrade.

The loads were simulated with plates for which the ratio of the loading plate diameter (D_p_) to the head diameter (D_h_) was 0.2, 0.4, 0.6, 0.8 and 1.0. Columns were modeled as single pieces and in a square grid with sides equal to 2D_h_ (replacement ratio α = 0.2), 1.5D_h_ (α = 0.35).

The discrete models consisted of 15 thousand to 75 thousand finite elements.

The influence of the models sizes on the horizontal and vertical displacements of columns and surrounding soils was analyzed. In all cases, these displacements were close to zero near the bottom and side edges of the models.

Examples of Group I and II models for columns made with 2D_h_ spacing and loaded with plates satisfying the condition D_p_/D_h_ = 1 are shown in Figure 8 and Figure 9.

In total, the results for 105 models were obtained and analyzed.

## 3. Results

The load–settlement relationship was determined for each model (Figure 10 and Figure 11). They formed the basis for determining the bearing capacity of the columns (Table 4) and the deformation modules of the strengthened soil (Figure 12 and Figure 13) according to Formula (1). According to Florkiewicz et al. [21], the bearing capacity of the columns was assumed to be equal to the load-causing settlement, equal to 10% of the diameter of their head. Results are presented for models in which the columns were formed at a spacing of two column-head diameters (L = 2D_h_). For clarity of the graphs, different line styles (e.g., continuous, dashed, etc.) were adopted for particular subgroups of columns (Ia–IIc) and the colors of curves selected so that, as the D_p_/D_h_ ratios decrease, the relationships are presented in increasingly lighter colors (from black to yellow).

For all subgroups of columns, the smallest settlements were obtained with the plates having the smallest diameter (D_p_/D_h_ = 0.2) and the largest settlements with the largest plates (D_p_/D_h_ = 1.0). This is due to the larger area of the plate base being drawn into the interaction zone, and this result is also obtained for homogeneous layers. However, the deformation moduli calculated on this basis according to Formula (1) for homogeneous layers will be similar. When the column–weak soil system is loaded, the weak surroundings of the column and the column’s underlying layer (in the case of Group II columns, this is soil of low strength and stiffness) are drawn into cooperation. This causes an increase in settlement and thus a decrease in the deformation modulus, which can be observed in Figure 12 and Figure 13. While for plates meeting the condition D_p_/D_h_ ≤ 0.4 in the initial load range (0–200 kPa), the obtained deformation moduli are close to the column modulus of elasticity (E = 60 MPa), for larger plates these values decrease even to approximately 20% of this modulus (e.g., Subgroup IIc, D_p_/D_h_ = 1.0, q = 0–200 kPa). In the remaining load range (q > 200 kPa), the final moduli obtained from testing with plates meeting the condition D_p_/D_h_ > 0.2 are clearly lower (1–8 MPa) than when using the smallest plate (approximately 20–30 MPa).

The value of the settlements is strongly influenced, first of all, by the shape of columns and their underlying layer; however, this depends on the diameter of the plates used. In the case of the smallest plates (D_p_/D_h_ = 0.2), the differences in settlements between individual column subgroups range from 36% (Group I) to 48% (Group II). For larger plates (D_p_/D_h_ ≥ 0.4), in the case of end bearing columns, the smallest settlements were obtained for the columns with the largest diameters in the lower parts of the columns (Subgroups Ib and Id), and the largest for the columns with the maximum diameter in the central part of the column (Ic) or with a constant diameter (Ia)—in this case, the differences in settlements were over 250% (e.g., for D_p_/D_h_ = 0.6). Similar behaviors were observed in the case of the floating columns—the smallest settlements were obtained when the diameter of the columns increased with depth (Subgroup IIb), and the largest in the opposite situation (Subgroup IIc). For this subgroup of columns, the differences in settlements reach approximately 450% (e.g., for D_p_/D_h_ = 1.0). These differences are translated into calculated deformation moduli (Figure 12 and Figure 13). When using plates with the same D_p_/D_h_ ratio, the differences may be up to 250% (e.g., for q = 400 kPa, Subgroups IIc and IIb, D_p_/D_h_ = 1.0).

The shape of the columns also affects the obtained limit stress. Among all the subgroups, the columns with diameters widening with depth (Subgroups Ib, Id and IIb) were characterized by the highest bearing capacity, while those with diameters decreasing with depth (IIc) with the lowest (Table 4). These differences are significant for plates meeting the condition D_p_/D_h_ ≥ 0.6; they reach 40–60%. It is worth noting that, despite the fact that the columns of Subgroup IIb were floated, they had a bearing capacity similar to that of the columns founded on the bearing layer. For plates with diameters satisfying the condition D_p_/D_h_ ≤ 0.4, high surface pressures (>1400 kPa) are necessary to obtain limit stress.

## 4. Discussion

The results of numerical analyses, presented in Section 3, in the form of load-test simulations, clearly show the significant influence of the shape (subgroup) of the columns on their settlement and bearing capacity. The shape of the columns, their length and how they are founded (end bearing or floating columns) influence the distribution of stresses produced during the load test. Figure 14, Figure 15 and Figure 16 show the distributions of the total horizontal and vertical stresses for the selected columns in Groups I (Figure 14 and Figure 15) and II (Figure 16), for the selected load level, i.e., q = 400 kPa (Groups I and II). The columns for which the smallest (Subgroups Ib, Id and IIb) and the largest (Subgroups Ia, Ic and IIc) settlements from each of the two groups were obtained were selected for analysis. For end bearing columns and with a diameter that increases with depth (Subgroup Ib) and with a maximum diameter at the center and the bottom of the column (Subgroups Ic and Id), the vertical stresses decrease with depth. Their value on the top of the bearing layer is about 141 kPa on average (about 35% of the test load value) for columns from Subgroup Ib, and about 172 and 249 kPa (about 43 and 62% of the test load value) for columns from Subgroups Id and Ic, respectively. For columns of constant diameter (Subgroup Ia) and the smallest length, the stresses transferred from the column to the bearing soil are approximately 377 kPa, and thus represent approximately 94% of the applied load. Therefore, the share of subsoil settlement under the column from Subgroup Ia in relation to the total settlement increases. This can be seen in Table 5, where the settlements of the layer underlying Column Ia constitute about 18% of the total column settlements. For the remaining subgroups of Group I, the settlement of the underlying layer does not exceed 10% of the total settlement. Column head settlements also depend on horizontal stresses transferred to the ground environment causing horizontal displacements of the column. Back in the 1970s, Hughes and Withers [22], based on the results of test loads, found that the column material in the upper parts of the columns moved laterally, and the strength of the surrounding soil had a great influence on the settlement and bearing capacity of the column. The horizontal stresses at the point of maximum horizontal displacement (Figure 17a,b) for columns from Subgroup Ia are about 21 kPa (Figure 14a), and are 40% higher than the horizontal stresses for columns of Subgroup Ib (Figure 14b) at about 15 kPa. This also results in increased settlements of columns of Subgroup Ia due to their higher horizontal displacements. For columns of Subgroup Ic, decreasing its diameter below half of its height increases the horizontal stress from this level (Figure 14c) and increases the zone of horizontal displacement and its values (Figure 17c).

The influence of the vertical stress distribution in the loaded system is particularly important in the case of floated columns (Group II), underlain by low-strength and high-deformability soils. In the case of columns of Subgroup IIb, the column diameter increasing with depth causes a decrease in vertical stresses transferred from the column base to the underlying soil. In the analyzed case, the mentioned vertical stresses constitute approximately 38% of the load applied on the head (approximately 152 kPa out of 400 kPa, Figure 16b). For the column of Subgroup IIc, with a diameter decreasing with depth, the same parameter is approximately 130% (approximately 521 kPa out of 400 kPa, Figure 16d), respectively. Therefore, the settlements of the layer underlying the column from Subgroup IIc constitute up to 50% of the total settlements of the column (Table 5). For the column of Subgroup IIb, the corresponding value is approximately 20%, which, in turn, is similar to the values obtained for end bearing columns (Table 5). It is also worth noting that, in case of the columns of Subgroup IIc, the influence of the presence of the weak layer under the column base is visible in the form of increased settlements of the layers under the base, when the load test is applied by using plates of a diameter meeting the condition D_p_/D_g_ ≥ 0.6 (Table 5). The shape of the Subgroup IIc columns is close to the wedge shape, which results in the formation of horizontal stresses on the column sides of up to 100% higher than those of Subgroup IIb columns (Figure 16a,c), and this, in turn, causes an increase in the vertical displacements of the columns.

In the case of a column of Subgroup Ib (Figure 18a), right at its side, the direction of the major component of the principal stress is parallel to it. The minor component has a small value. In the case of a column of Subgroup Ic, such a situation occurs only in the upper part of the column (Figure 18b), near the maximum diameter and below which there are already two components of principal stresses, i.e., vertical and horizontal (Figure 18b), with similar values. For columns of Subgroup IIc, the principal stress components are arranged along the entire length of the column side surface in the vertical and horizontal directions (Figure 18c) and have similar values. For columns of Subgroups Ic and IIc, this stress distribution increases the settlement of the column under the load test.

The calculations also allowed us to determine the effect of column spacing on the results of the load tests. This effect depends on the diameter of the loading plate and the type of column subgroup. Table 6 shows the percentage decrease in settlement of columns made at a spacing of one and a half times the column head diameter (1.5D_h_) and twice the head diameter (2.0D_h_) compared to the settlement of a single column over the load range up to the limits loads in Table 4.

For the smallest of the plates used (D_p_/D_h_), the adjacent columns reduce settlement to about 8% with a column spacing of 1.5D_h_. For greater spacing (2.0D_h_), there was no notable effect of adjacent columns on the results (<1%). As the diameter of the loading plates increases, the influence of the adjacent columns also increases. For each of the subgroups, the greatest effect of adjacent columns, for all plates used, was observed at a smaller column spacing (1.5D_h_).

Among Group I columns, the greatest influence of adjacent columns was observed for Subgroup Ib. The percentage reduction of settlements in this case was 36% (for 2D_h_ spacing) and 52% (for 1.5D_h_). For Group II, the greatest influence of adjacent columns was obtained for columns in Subgroup IIc, i.e., 47% (for 1.5D_h_ spacing) and 23% (for 2.0D_h_). The smallest influence of adjacent columns, on the other hand, was recorded for Subgroups Ia and IIa, and thus for columns of constant diameter. For the 1.5D_h_ spacing, the percent reduction in settlement did not exceed approximately 27% (Ia) and 25% (IIa). For columns made at the spacing of 2.0D_h_, the corresponding values were 13% (Ia) and 8% (IIa), respectively.

## 5. Conclusions

The calculations, results and their analyses showed that the results of the load tests of columns formed by using the dynamic replacement method were affected by the loading plate diameter, type of column and the occurrence of adjacent columns. The larger the diameter of the plate used, the larger the area drawn into the interaction zone. The shape of the columns and their length influenced the distribution of the horizontal and vertical stresses during the load tests, which, in combination with how the columns are founded, may cause their excessive vertical displacement. Closer proximity to other columns, in turn, limits the settlement of the column.

The analysis showed that the evaluation of the columns under the load test is inconclusive when based only on the calculated deformation modulus (according to Formula (1). This may lead to erroneous conclusions, e.g., disqualification of a column tested with plates of diameters close to the head of the column (insufficient deformation modulus obtained) or acceptance of a floated column (insufficient diameter of the loading plate). Therefore, the interpretation of the results of the test loads should be based on advanced numerical analysis by means of the finite element method, using appropriate constitutive models, where, in addition to the knowledge of the soil and its parameters, it is necessary to recognize the shape of the column (by excavations [10] or geophysical tests [23]), how the base is founded (by excavations, geophysical tests or dynamic probing test (DPH—Dynamic Penetrometer Heavy and DPSH—Dynamic Penetrometer Super Heavy) and the occurrence of adjacent columns. In this case, the use of semi-inverse FEM analysis allowed the determination of the parameters of the columns and their comparison with the parameters adopted at the design stage.

## Figures and Tables

**Figure 1 sensors-21-04868-f001:**
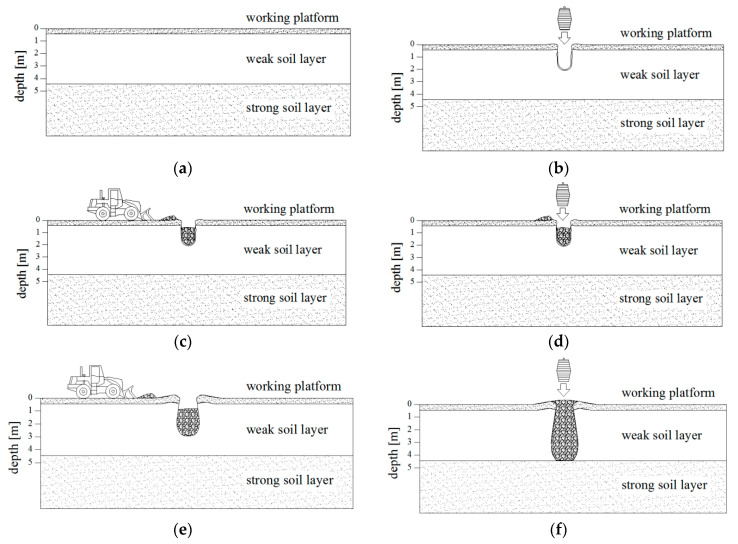
Process of dynamic replacement column formation: (**a**) construction of working platform, (**b**) drop of tamper and crater creation, (**c**) crater backfill and (**d**,**e**) drop of tamper and crater backfill, (**f**) dynamic replacement column.

**Figure 2 sensors-21-04868-f002:**
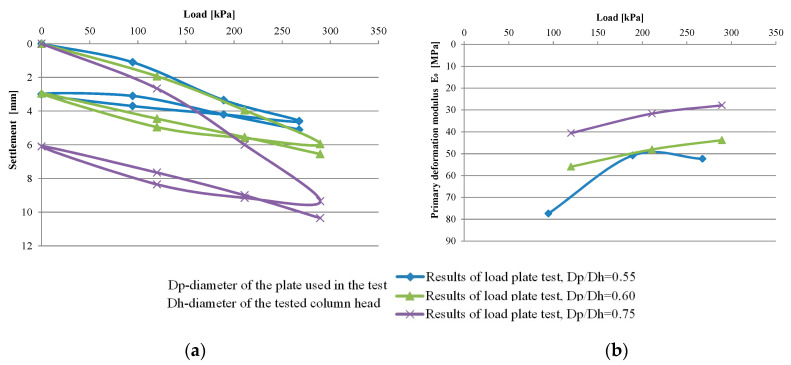
Results of load plate tests [10]: (**a**) load–settlement relationship and (**b**) values of primary deformation modulus.

**Figure 3 sensors-21-04868-f003:**
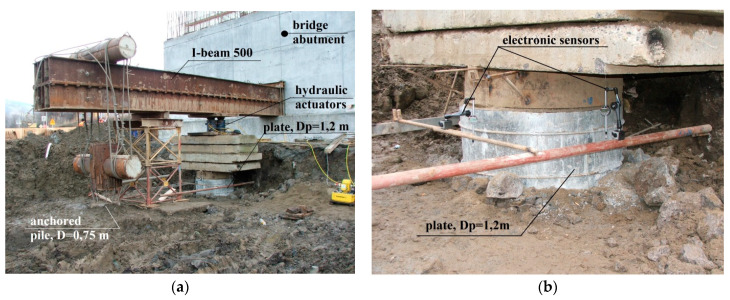
Trial load station to ultimate bearing capacity of column test: (**a**) general view and (**b**) plate and electronic sensors.

**Figure 4 sensors-21-04868-f004:**
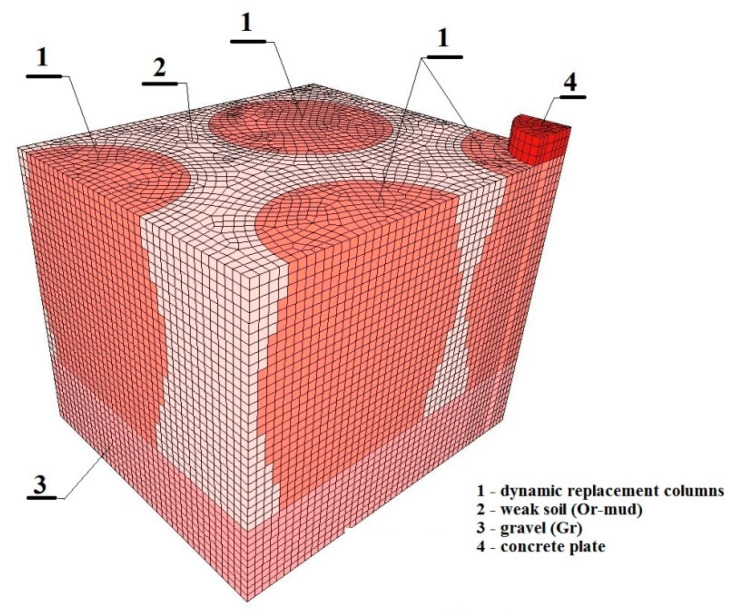
Discrete FEM model of ground improved by dynamic replacement method [10].

**Figure 5 sensors-21-04868-f005:**
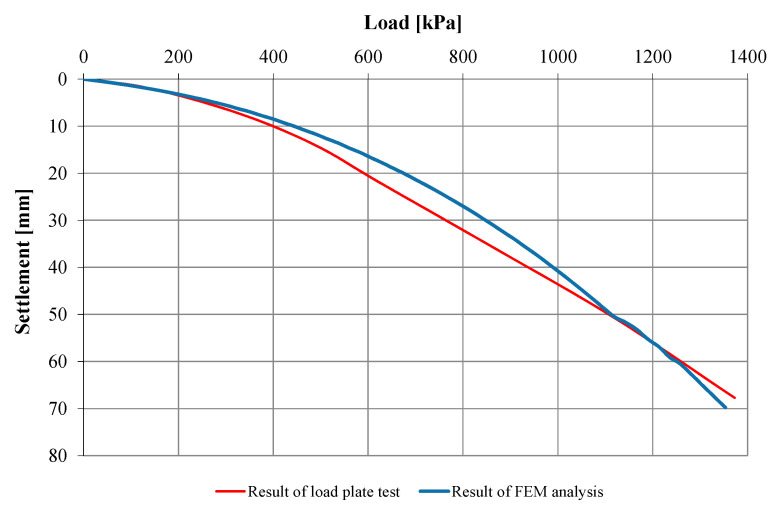
Results of semi-inverse analysis [10].

**Figure 6 sensors-21-04868-f006:**
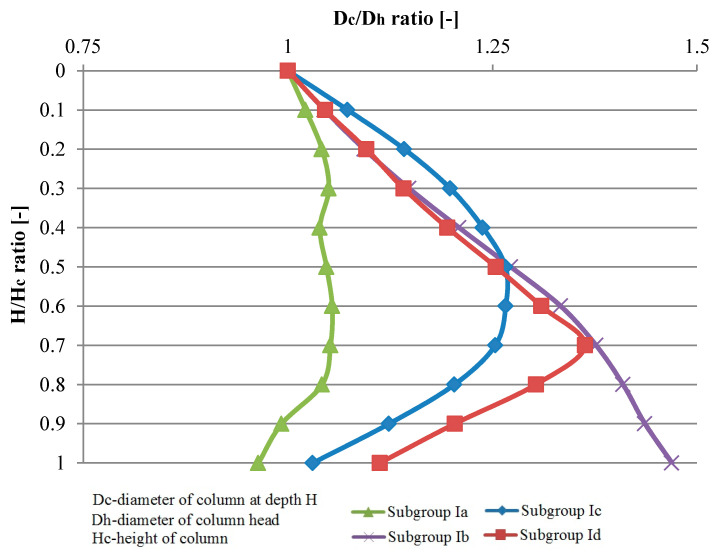
Columns diameter variability (mean value) on their length—Group I.

**Figure 7 sensors-21-04868-f007:**
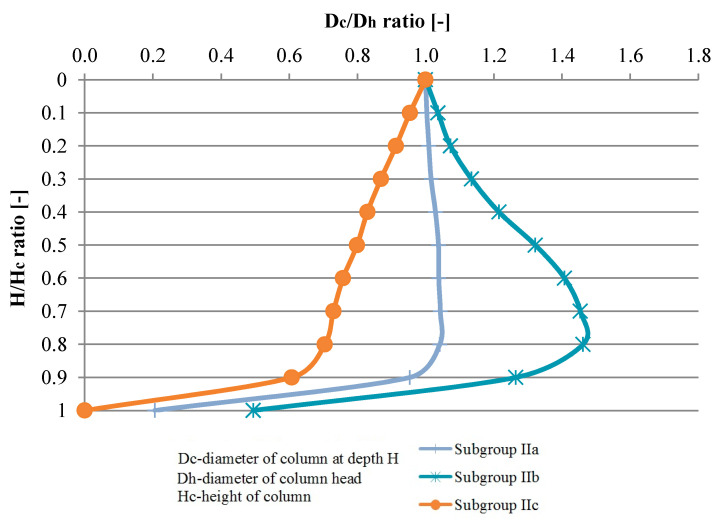
Columns diameter variability (mean value) on their length—Group II.

**Figure 8 sensors-21-04868-f008:**
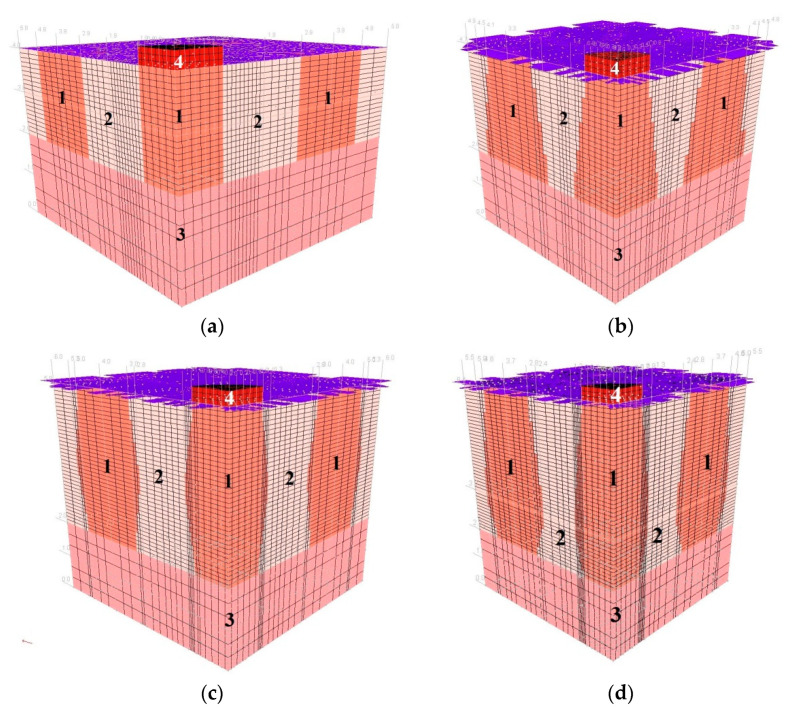
Discrete models of ground improved by dynamic replacement—Subgroup I, L = 2D_h_: (**a**) Ia, (**b**) Ib, (**c**) Ic and (**d**) Id.

**Figure 9 sensors-21-04868-f009:**
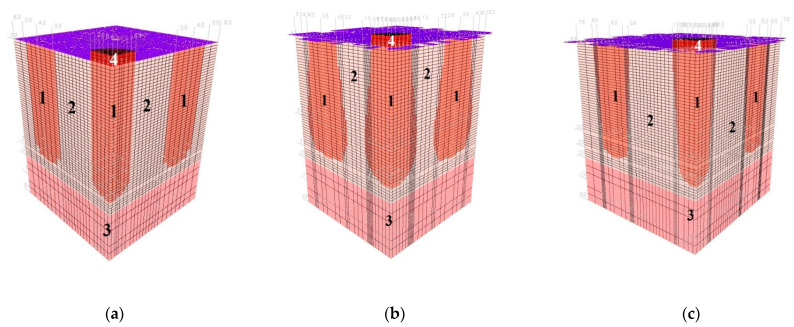
Discrete models of ground improved by dynamic replacement—Subgroup II, L = 2D_h_: (**a**) IIa, (**b**) IIb and (**c**) IIc.

**Figure 10 sensors-21-04868-f010:**
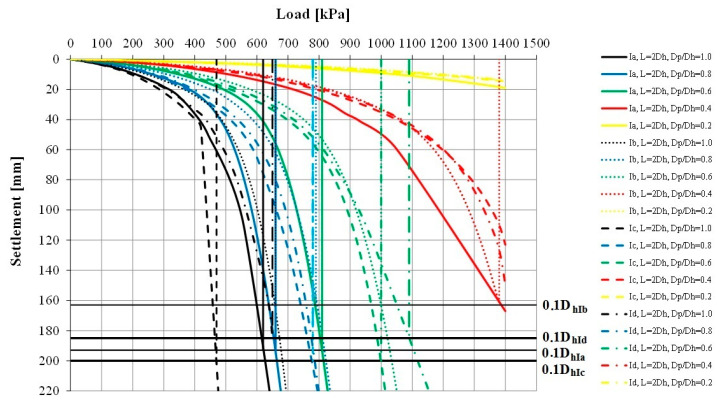
“Load–settlement” relationship—Columns Group I.

**Figure 11 sensors-21-04868-f011:**
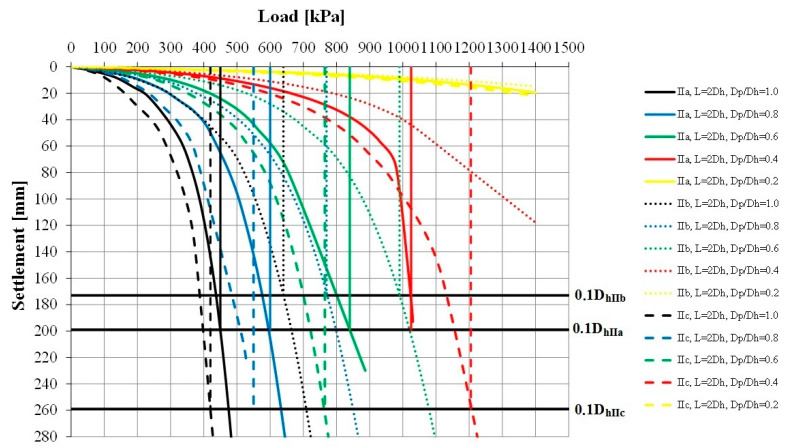
“Load–settlement” relationship—Columns Group II.

**Figure 12 sensors-21-04868-f012:**
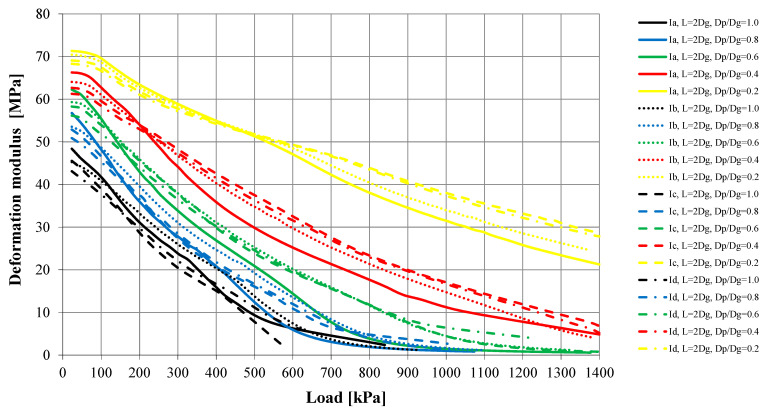
Calculated deformation modulus (E) of improvement ground—Columns Group I.

**Figure 13 sensors-21-04868-f013:**
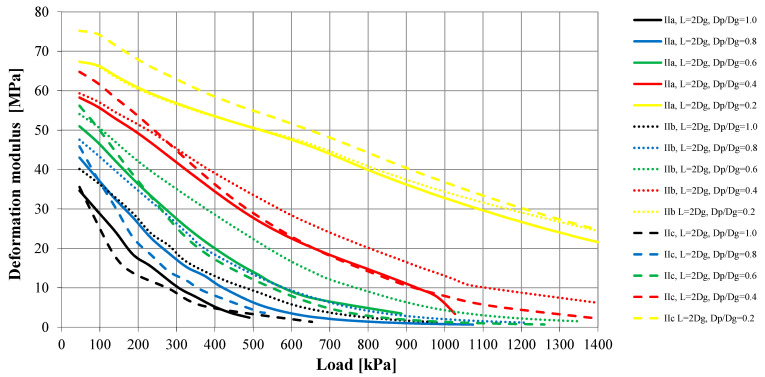
Calculated deformation modulus (E) of improvement ground—Columns Group II.

**Figure 14 sensors-21-04868-f014:**
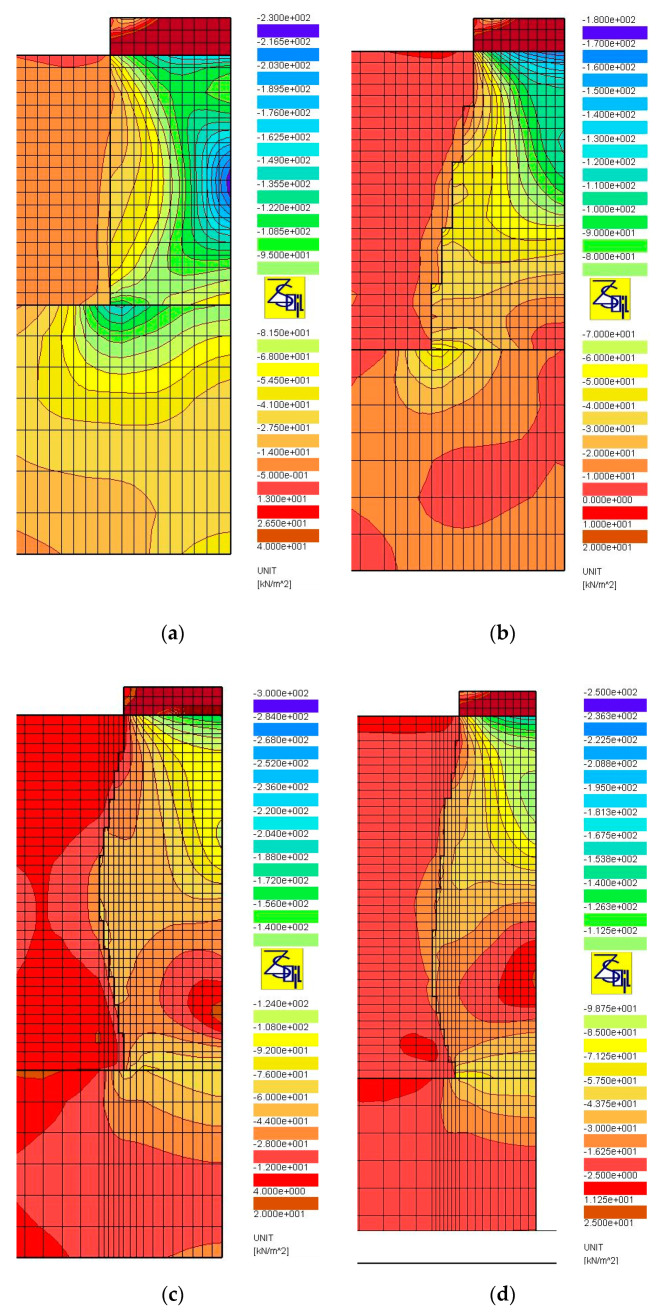
Maps of horizontal total stress, q = 400 kPa, D_p_/D_h_ = 1.0: (**a**) Subgroups Ia, (**b**) Ib, (**c**) Ic and (**d**) Id.

**Figure 15 sensors-21-04868-f015:**
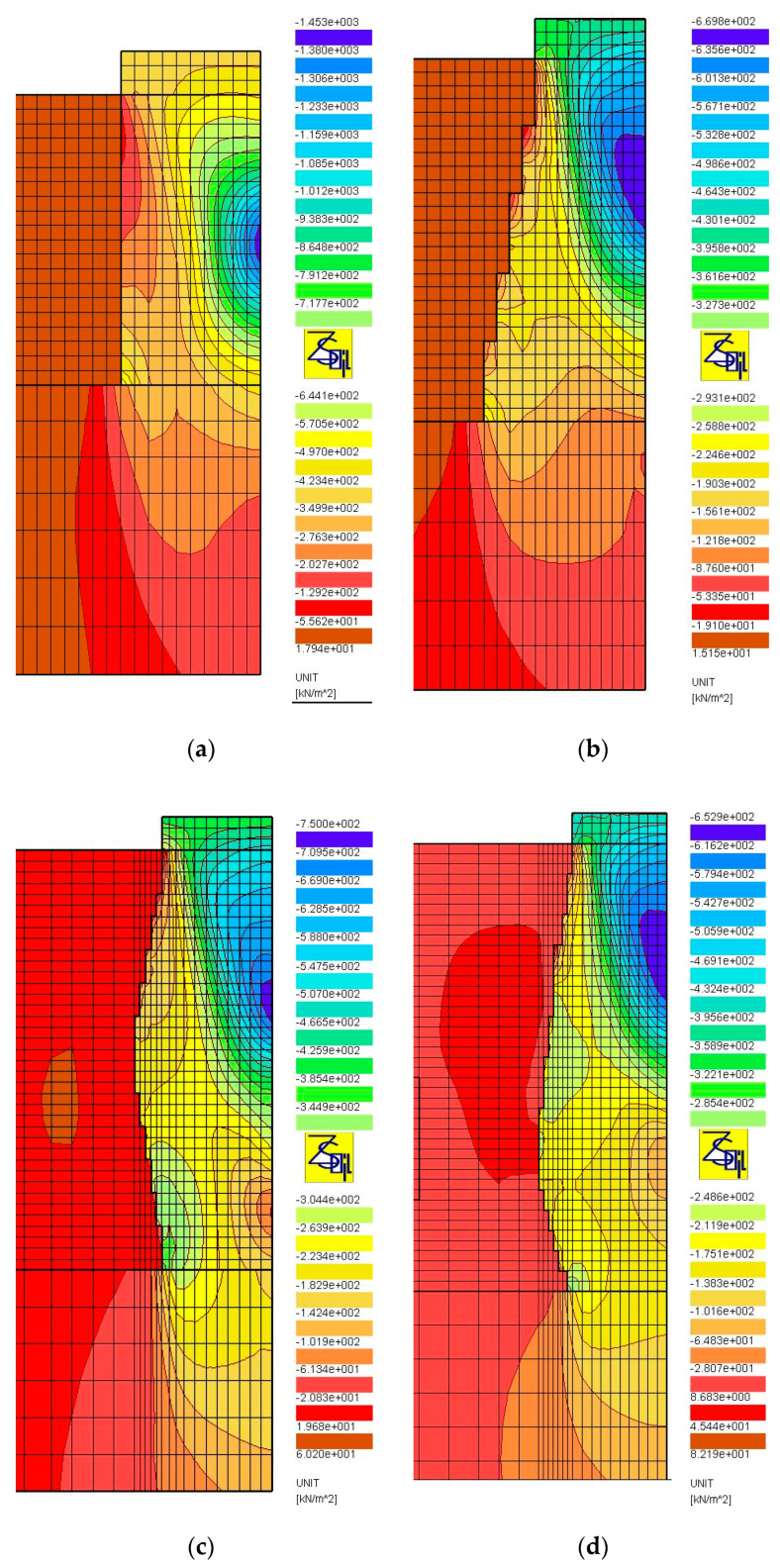
Maps of vertical total stress, q = 400 kPa, D_p_/D_h_ = 1.0: (**a**) Subgroups Ia, (**b**) Ib, (**c**) Ic and (**d**) Id.

**Figure 16 sensors-21-04868-f016:**
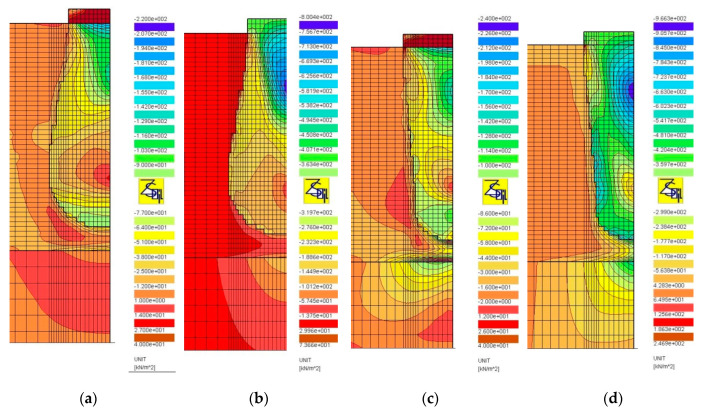
Maps of total stress, q = 400 kPa, D_p_/D_h_ = 1.0: (**a**) horizontal, Subgroup IIb; (**b**) vertical, Subgroup IIb; (**c**) horizontal, Subgroup IIc; (**d**) vertical, Subgroup IIc.

**Figure 17 sensors-21-04868-f017:**
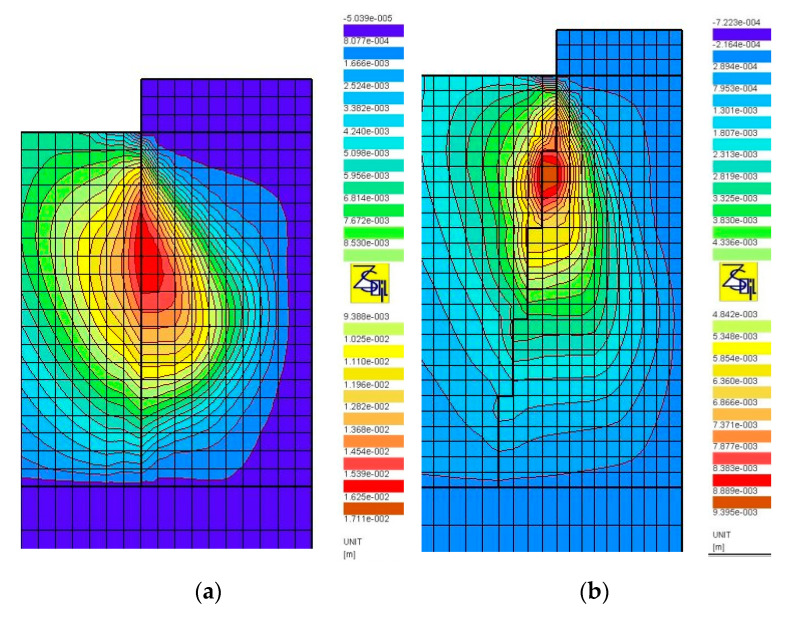
Maps of horizontal displacement, q = 400 kPa, D_p_/D_h_ = 1.0: (**a**) Subgroups Ia, (**b**) Ib, (**c**) Ic and (**d**) Id.

**Figure 18 sensors-21-04868-f018:**
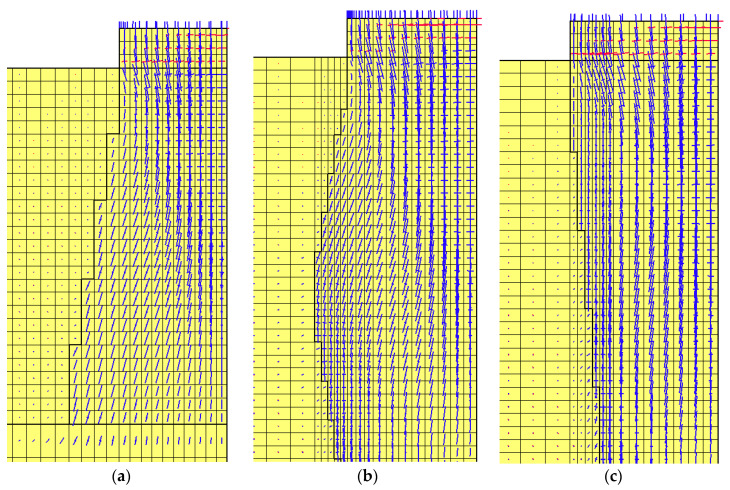
Maps of principal stresses directions, D_p_/D_h_ = 1.0, q = 400 kPa: (**a**) Subgroups Ib, (**b**) Ic and (**c**) IIc.

**Table 1 sensors-21-04868-t001:** Physical and mechanical characteristics of the weak soil (clayey mud) surrounding the column.

Organic Content I_om_ (%)	Plastic/Liquid Limit w_p_/w_L_ (%)	Consistency Index Ic (-)	Bulk Unit Weight γ (kN/m^3^)	Natural Water Content w_n_ (%)	Initial Void Ratio e_0_ (-)	Permeability Coefficient k_v_ (m/s)	Modulus of Primary Compress-Ion (MPa)
3.3	25.2/55.8	0.66	18.4	35	0.84	1.25 × 10^−9^	2.3

**Table 2 sensors-21-04868-t002:** Constitutive models and parameters.

Material Zone	Constitutive Model	Parameters and Initial Values
Columns (1)	Mohr–Coulomb, an elastic–perfectly plastic model	E = 60 MPa, υ = 0.25, ϕ = 46°, c = 5 kPa, ψ = 20°, k_x_ = k_y_ = k_z_ = 1 × 10^−2^ m/s
Clayey mud (2)	Modified Cam-Clay, an elastic isotropic hardening model	λ = 0.053, κ = 0.0028, e_0_ = 0.84, M = 1.48, OCR = 1.1, κ = 0.3, k_x_ = k_y_ = k_z_ = 1.25 × 10^−9^ m/s
Gravel (3)	Mohr–Coulomb, an elastic–perfectly plastic model	E = 140 MPa, υ = 0.25, ϕ = 38.5°, c = 0 kPa, ψ = 8.5ψ, k_x_ = k_y_ = k_z_ = 1 × 10^−2^ m/s
Concrete plate (4)	Linear-elastic	E = 27° GPa, υ = 0.2.

Note: E—Young’s modulus, υ—Poisson’s ratio, λ—slope of isotropic normal consolidation line, κ—slope of isotropic swelling line, M—slope of critical state line, OCR—overconsolidation ratio, k—permeability coefficient, e_0_—initial void ratio, ϕ—angle of internal friction, c—cohesion, ψ—dilatancy angle.

**Table 3 sensors-21-04868-t003:** Classification of the investigated dynamic replacement columns.

Group	Type (Subgroup)	Columns
I. End bearing columns (flat columns bases) Total of 35 columns	Ia—cylindrical columns; strengthened soils thickness: 1–2 m; D_hm_ = 1.93 m (D_hm_—column head diameter (mean value))	Ia_1_–Ia_3_, 3 columns
	Ib—columns with a diameter increasing with the depth; strengthened soils thickness: 2–2.7 m; D_hm_ = 1.63 m	Ib_1_–Ib_3_, 3 columns
	Ic—columns with the maximum diameter near the middle of length; strengthened soils thickness: 2.7–3.8 m; D_hm_ = 2.0 m	Ic_1_–Ic_23_, 23 columns
	Id—columns with a diameter increased at the bottom; strengthened soils thickness: 3.8–4.4 m; D_hm_ = 1.85 m	Id_1_–Id_6_, 6 columns
II. Floating columns (semi-circular columns bases) Total of 30 columns	IIa—approximately cylindrical columns; columns length: 3.0–4.5 m; D_hm_ = 1.99 m	IIa_1_–IIa_10_, 10 columns
	IIb—enlarged base columns; columns length: 3.0–4.2 m; D_hm_ = 1.73 m	IIb_1_–IIb_10_, 10 columns
	IIc—enlarged head columns; columns length: 1.9–4.4 m; D_hm_ = 2.59 m	IIc_1_–IIc_10_, 10 columns

**Table 4 sensors-21-04868-t004:** Bearing capacity of loaded columns.

Bearing Capacity of Columns (kPa)							
Subgroup D_p_/D_h_ (-)	Ia	Ib	Ic	Id	IIa	IIb	IIc
0.2	>1400	>1400	>1400	>1400	>1400	>1400	>1400
0.4	>1400	1380	>1400	>1400	1025	>1400	1205
0.6	810	1000	1000	1090	840	990	765
0.8	660	790	780	780	600	770	550
1.0	620	660	470	650	450	640	420

**Table 5 sensors-21-04868-t005:** Influence of the columns underlying layer on total settlement.

Base Column Settlement to Total Column Settlement Ratio (%)							
Subgroup D_p_/D_h_ (-)	Ia	Ib	Ic	Id	IIa	IIb	IIc
0.2	4.9	2.9	2.2	1.6	2.3	2.2	2.9
0.4	9.1	5.3	4.0	2.9	3.9	3.9	10.8
0.6	12.8	7.4	5.6	4.0	9.7	8.0	24.9
0.8	15.8	8.9	6.8	4.8	18.2	14.3	37.9
1.0	17.7	9.7	8.4	5.3	25.1	20.4	49.9

**Table 6 sensors-21-04868-t006:** Influence of the spacing of columns on the results of test loads.

Reduction of Settlement (%)														
	Ia		Ib		Ic		Id		IIa		IIb		IIc	
	1.5D_h_	2D_h_	1.5D_h_	2D_h_	1.5D_h_	2D_h_	1.5D_h_	2D_h_	1.5D_h_	2D_h_	1.5D_h_	2D_h_	1.5D_h_	2D_h_
0.2	7.8	<1.0	3.1	<1.0	1.4	<1.0	<1.0	<1.0	<1.0	<1.0	2.9	<1.0	5.2	<1.0
0.4	15.5	4.2	36.7	25.9	28.7	6.0	20.1	10.1	25.3	6.8	32.5	<1.0	21.6	4.7
0.6	27.4	13.0	28.0	13.9	20.4	5.6	16.9	10.0	24.9	7.9	30.3	8.3	24.9	10.6
0.8	18.0	2.6	51.8	32.0	26.5	6.6	12.9	6.5	23.0	7.0	30.6	9.2	47.2	12.9
1.0	15.8	5.0	50.6	36.0	21.1	5.3	27.7	11.5	20.4	3.0	29.7	10.0	44.1	23.1

## Data Availability

The data presented in this study are available on request from the corresponding author. The data are not publicly available due to large file size.

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
