# Peer review of "Influence of Load Plates Diameters, Shapes of Columns and Columns Spacing on Results of Load Plate Tests of Columns Formed by Dynamic Replacement"

_sensors, 2021, doi:10.3390/s21144868_

Round 1
Reviewer 1 Report
Please find the attached.

Author Response
Thank you very much for your review of the article. I agree with most of the comments and made corrections. Details in uploaded files. All changes are highlighted in yellow in the text.

Reviewer 2 Report
The paper shows the numerical analysis for the soil improvement method using columns. It would be better in terms of engineering practice if the authors analyze the efficiency of the method. For example, what is the optimal value of the diameter and length of the columns to have an efficient and economical design?
Author Response

(The authors gave the same response as above.)

Reviewer 3 Report
The testing methods are insufficient to determine the soil column interaction. Soil parameters are not obtained from in situ testes, the laboratory tests are not presented. Please refer to geotechnical standard tests when determining the parameters of the soil in natural state and improved soil parameters.
The parameters used in analysis do not correspond to the usual soils parameters considered in geotechnical analysis (see Table 2), so the results might be affected by this aspect.
Please reconsider the testing procedure for the columns influence. Please consider as references for the paper recent tests and papers about this subject.
Author Response

(The authors gave the same response as above.)

Round 2
Reviewer 3 Report
The numerical study is extensive , but the results are common sense.
The conclusion do not present the relevance of the study.